# Analysis of Factors Associated with Highway Personal Car and Truck Run-Off-Road Crashes: Decision Tree and Mixed Logit Model with Heterogeneity in Means and Variances Approaches

**Thanapong Champahom [1], Panuwat Wisutwattanasak [2], Chamroeun Se [2], Chinnakrit Banyong [3], Sajjakaj Jomnonkwao [3,*] and Vatanavongs Ratanavaraha [3]**

1   Department of Management, Faculty of Business Administration, Rajamangala University of Technology Isan, Nakhon Ratchasima 30000, Thailand; thanapong.ch@rmuti.ac.th
2   Institute of Research and Development, Suranaree University of Technology, Nakhon Ratchasima 30000, Thailand; panuwat.w@g.sut.ac.th (P.W.); chamroeun.s@g.sut.ac.th (C.S.)
3   School of Transportation Engineering, Institute of Engineering, Suranaree University of Technology, Nakhon Ratchasima 30000, Thailand; d6500733@g.sut.ac.th (C.B.); vatanavongs@g.sut.ac.th (V.R.)
*   Correspondence: sajjakaj@g.sut.ac.th

**Abstract:** Among several approaches to analyzing crash research, the use of machine learning and econometric analysis has found potential in the analysis. This study aims to empirically examine factors influencing the single-vehicle crash for personal cars and trucks using decision trees (DT) and mixed binary logit with heterogeneity in means and variances (RPBLHMV) and compare model accuracy. The data in this study were obtained from the Department of Highway during 2011–2017, and the results indicated that the RPBLHMV was superior due to its higher overall prediction accuracy, sensitivity, and specificity values when compared to the DT model. According to the RPBLHMV results, car models showed that injury severity was associated with driver gender, seat belt, mount the island, defect equipment, and safety equipment. For the truck model, it was found that crashes located at intersections or medians, mounts on the island, and safety equipment have a significant influence on injury severity. DT results also showed that running off-road and hitting safety equipment can reduce the risk of death for car and truck drivers. This finding can illustrate the difference causing the dependent variable in each model. The RPBLHMV showed the ability to capture random parameters and unobserved heterogeneity. But DT can be easily used to provide variable importance and show which factor has the most significance by sequencing. Each model has advantages and disadvantages. The study findings can give relevant authorities choices for measures and policy improvement based on two analysis methods in accordance with their policy design. Therefore, whether advocating road safety or improving policy measures, the use of appropriate methods can increase operational efficiency.

**Keywords:** data driven; econometric analysis; model accuracy; sensitivity; specificity

## 1. Introduction

Thailand, classified as a middle-income country, faces a significant number of serious crashes. As of 2018, the fatality rate stood at 32.7 per 100,000 people, ranking it eighth globally [1]. An analysis of data from the Highway Crash Information Management System [2] reveals that between 2011 and 2017, Thailand experienced the highest proportion of run-off-road crashes, accounting for approximately 52% (Figure 1). The current focus on addressing road accidents has led to increasing interest in the use of automated vehicles (AVs) as a potential solution. Scholars have highlighted the advantageous features of AVs, such as driving assistance systems and advanced sensors, which contribute to their ability to prevent accidents [3,4]. Moreover, the rise of AVs aligns with the growing popularity of electric vehicles, leading to not only a reduction in road risks but also the promotion of a

greener environment and the advancement of industry 4.0 technology [5,6]. This aligns with the Sustainable Development Goals (SDGs) established to foster sustainability. However, it is important to acknowledge that before fully embracing these automatic and green industries, developing countries must first tackle the immediate issue of road accidents. Researchers have studied factors that affect the severity of collision crashes using various methods to solve road crash problems, as well as the correlation between driver factors and crash occurrence [7]. Figure 2 provides valuable insights into the fatality rate associated with crashes involving personal cars and trucks. Although these vehicles account for 46% of all crashes [1], it is important to note that they represent the medium- and large-sized vehicle categories. Literature findings from sources such as [8–10] further support the notion that car and truck-related crashes tend to result in more severe injuries and cause greater damage to both private and public properties compared to smaller groups of road users like pedestrians and motorcyclists. Additionally, it is worth highlighting that among the different types of crashes, single-vehicle incidents hold the highest proportion [11,12]. This information underscores the significance of considering the factors contributing to and consequences of single-vehicle crashes in efforts to enhance road safety [13].

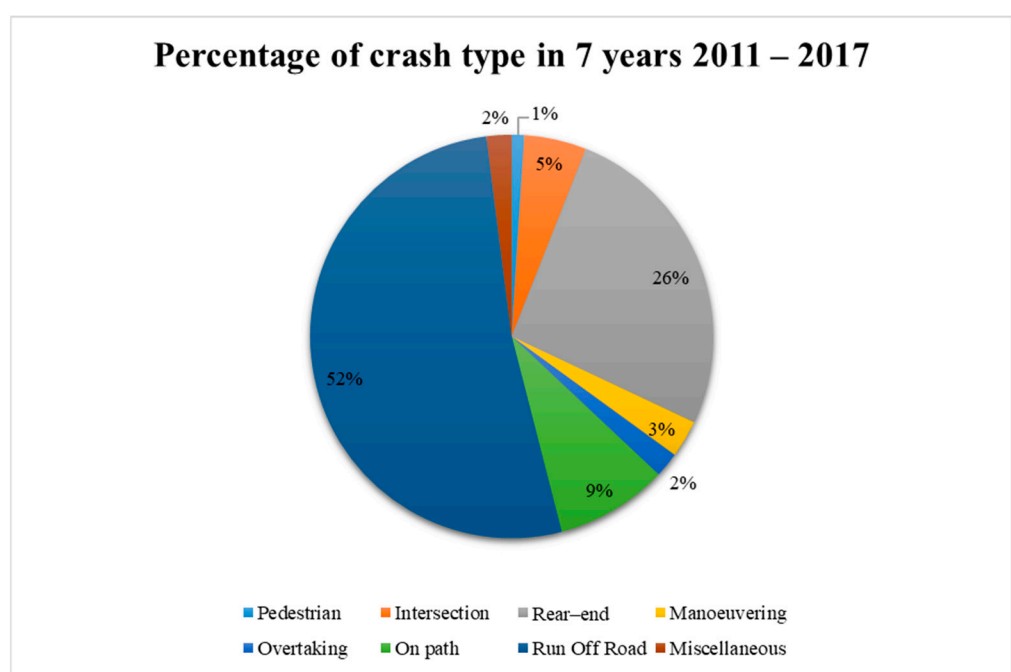

**Figure 1.** Crash type percentage of 7 years (2011–2017) of crash records in Thailand.

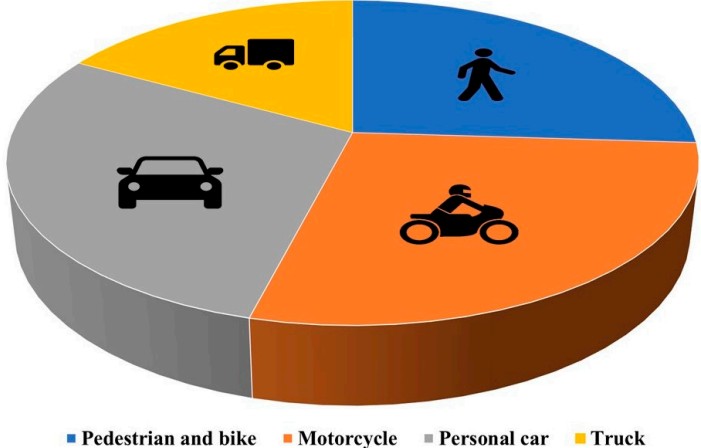

**Figure 2.** Crash fatality percentage classified by vehicle type.

Nowadays, numerous types of machine learning (ML) are applied in the study of crash severity; the decision tree (DT) is the method applied to algorithms' arrangements to recognize the proportion of data based on determining variables. Thus, the researcher can use appropriate data to analyze complicated independent data [14,15]. The use of DT as an ML model offers a distinct advantage over other models, such as Artificial Neural Networks. Specifically, DT has the ability to determine the order of influence of independent variables on dependent variables through nodes and branches. In contrast, other ML models that utilize Blackbox algorithms cannot reveal the priority of factors. It is worth noting that DT is not able to improve model depth or increase accuracy when there is unobserved heterogeneity, unlike some other ML models. Nevertheless, the ability to obtain the order of important variables and a classification tree structure allows for meaningful comparisons with other econometric models, such as the logit model. According to the literature, it has been applied to crash analysis and predicted non-injury crashes in Malaysia [16] and crashes at the crosspoint between roads and railways in the United States [15]. In Thailand, the DT model was also used for analyzing rear-end collisions on Thai highways [17]. Nevertheless, if we can compare the unique results of DT with another method, it will give us different aspects.

Additionally, the logit model is a widely used method for predicting crashes and illustrates the comparison of the characteristics of different severity levels [18]. For example, Champahom, et al. [18] compared the severity of crash-related incidents on urban and rural roads in Thailand, Chen, et al. [19] studied the severity of injuries among truck drivers, and Huang, et al. [20] studied the severity of driver injury and vehicle damage in traffic congestion at intersections in Singapore [19,21]. The traditional logit model, on the other hand, appears to be an inefficient method for analyzing data with a large number of variables. Nowadays, scholars have established a random parameter model that can capture the variation and complexity of the model [22–24]; the extension of logit can describe the relationship between the fixed parameters of the model and can also explain the model's variation. Moreover, researchers recently uncovered the concept of unobserved heterogeneity (in means and variances), which can apply to random parameters (probit) logit model in the analysis of traffic crash injury severity [25,26]; this represents a hidden influence (layer 2) that can affect the direction of random parameters of the model, this method could influence model complexity.

The aforementioned methods have been used in crash analysis and are based on the researcher's objectives, which vary according to the research context and objectives, as each model has a different form of operation (algorithm). However, the proposed studies of DT and RPBLHMV analysis have not yet been discovered. Due to the quite different functions of the model, a study that can compare the significant results and the model's performance of each method could reveal the model's advantages and disadvantages and lead to efficient use [27].

This study foresees the potential for predicting the crash injury outcome of both models and the explanatory significance of significant results. So, the objective is to compare the crash data analysis from contributing factors, such as vehicle factors, driver factors, and road and environmental factors, to determine how they affect crash severity (fatal and non-fatal). To achieve this objective, this study mainly used two different techniques: The first is the data-driven technique (DT), and the second is econometric analysis (random parameters binary logit model with heterogeneity in means and variances: RPBLHMV) to analyze factors associated with drivers' injury severity among personal car and truck run-off-road crashes. Past studies have confirmed that both methods have potential in crash analysis and revealed the associating factors, resulting in plans, measures, or policies that will help reduce road crashes. In addition to affecting factors, this study wants to compare the model accuracy (predicting outcome) between the DT and RPBLHMV models, including overall accuracy, sensitivity, and specificity. The contribution of the study is that it enables readers to understand the advantages and disadvantages of each model and then select the appropriate method for analyzing run-off-road crashes on highways (among

personal car and truck drivers). In addition, these findings can give relevant authorities the contributing factors for establishing policies and measures to reduce the severity of run-off-road crashes. Good mitigation can result in both injury-related and property damage reduction.

## 2. Materials and Methods

### 2.1. Data Collection and Descriptive Statistics

This study used the crash database from the Department of Highways of Thailand, which comprised two parts: (1) data on highway crashes in Thailand from 2011 to 2017, as reported by police officers and recorded in the Highway Crash Information System Management (HAIMS), comprising causes, severity, driver characteristics, crash characteristics, vehicle characteristics, road characteristics, and environmental context. This dataset was screened for only personal car (total of 3448 cases) and truck (including at least 6-wheeled vehicles; a total of 1375 cases) crashes related to run-off-road collisions. In addition, collisions are categorized into two levels of severity: fatal (severe injury or fatality) and non-fatal (property-damaged only or minor injury) injuries. Variables are coded and described in Table 1.

**Table 1.** Summary statistic of single-vehicle crash data.

| | Variable | Description | Car (*n* = 3448) | | Truck (*n* = 1375) | |
|---|---|---|---|---|---|---|
| | | | **Mean** | **SD** | **Mean** | **SD** |
| Y | SEVERITY | 1 if severe or fatal injury; 0 PDO or minor injury | 0.283 | 0.450 | 0.253 | 0.435 |
| | AGE_26_35 | 1 if aged 26 to 35; 0 otherwise | 0.365 | 0.481 | 0.333 | 0.471 |
| | AGE_36_45 | 1 if aged 36 to 45; 0 otherwise | 0.221 | 0.415 | 0.320 | 0.467 |
| | AGE_46_55 | 1 if aged 46 to 55; 0 otherwise | 0.126 | 0.332 | 0.184 | 0.388 |
| | AGE_56_UP | 1 if the driver's age is more than 55; 0 otherwise | 0.084 | 0.277 | 0.058 | 0.234 |
| | MALE | 1 if male drivers; 0 otherwise | 0.773 | 0.419 | 0.991 | 0.093 |
| | SAF_EQ | 1 if driver uses seatbelt; 0 otherwise | 0.409 | 0.492 | 0.362 | 0.481 |
| | ALCOHOL | 1 if driver is under effect of alcohol; 0 otherwise | 0.017 | 0.131 | 0.007 | 0.085 |
| | EXEED_SPEED | 1 if driver exceeds speed limit; 0 otherwise | 0.808 | 0.394 | 0.703 | 0.457 |
| | FALL_ASLEEP | 1 if driver falls asleep while driving; 0 otherwise | 0.119 | 0.324 | 0.140 | 0.347 |
| | CONSTRUCT | 1 if crash occurs at area of road maintenance (or construction); 0 otherwise | 0.028 | 0.166 | 0.026 | 0.160 |
| | ASPHALT | 1 if pavement type is asphalt; 0 otherwise | 0.912 | 0.284 | 0.933 | 0.25 |
| | VERTICAL | 1 if crash occurs on the graded road section; 0 otherwise | 0.086 | 0.280 | 0.199 | 0.400 |
| | INTERSECTION | 1 if crash occurs within intersection; 0 otherwise | 0.071 | 0.256 | 0.085 | 0.279 |
| | U_TURN | 1 if crash occurs within U-turn (opened median); 0 otherwise | 0.099 | 0.298 | 0.056 | 0.230 |
| | COMMUNITY | 1 if crash occurs within community area; 0 otherwise | 0.010 | 0.100 | 0.009 | 0.093 |
| | NO_MEDIAN | 1 if crash occurs on road without median; 0 otherwise | 0.267 | 0.443 | 0.351 | 0.478 |
| | PAINTED | 1 if crash occurs on road with painted median; 0 otherwise | 0.051 | 0.22 | 0.031 | 0.172 |
| | RAISED | 1 if crash occurs on road with raised median; 0 otherwise | 0.280 | 0.449 | 0.181 | 0.385 |
| | DEPRESSED | 1 if crash occurs on road with depressed median; 0 otherwise | 0.345 | 0.475 | 0.363 | 0.481 |
| | BARRIER | 1 if crash occurs on road with barrier median; 0 otherwise | 0.050 | 0.217 | 0.069 | 0.254 |
| | MOUNT_ISLAND | 1 if the vehicle mounted the traffic island; 0 otherwise | 0.248 | 0.432 | 0.162 | 0.369 |
| | PASS_IN_FRONT | 1 if crash passes in front of car; 0 otherwise | 0.020 | 0.139 | 0.033 | 0.178 |
| | DEFECT_CAR | 1 if crash occurs because of defective car equipment; 0 otherwise | 0.012 | 0.110 | 0.066 | 0.247 |
| | WET_SURFACE | 1 if crash occurs on wet road; 0 otherwise | 0.155 | 0.362 | 0.199 | 0.400 |
| | DIRTY_SURFACE | 1 if crash occurs on wavy or dirty road; 0 otherwise | 0.004 | 0.061 | 0.004 | 0.066 |
| | WEATHER | 1 if crash occurs during rain, dust, or fog; 0 otherwise | 0.170 | 0.375 | 0.213 | 0.410 |
| | NIGHT | 1 if crash occurs during nighttime; 0 otherwise | 0.503 | 0.500 | 0.417 | 0.493 |
| | OFF_STR | 1 if cause of crash is being run off-road on a straight; 0 otherwise | 0.140 | 0.347 | 0.093 | 0.291 |
| | OFF_STR_HIT | 1 if cause of crash is being run off-road on a straight and striking safety equipment; 0 otherwise | 0.323 | 0.468 | 0.330 | 0.470 |
| | OFF_CUR | 1 if cause of crash is being run off-road on curve; 0 otherwise | 0.057 | 0.232 | 0.079 | 0.270 |
| | OFF_CUR_HIT | 1 if cause of crash is being run off-road on curve and striking safety equipment; 0 otherwise | 0.191 | 0.393 | 0.281 | 0.45 |

Note: SD = standard deviation; PDO = property-damaged only.

To avoid multicollinearity among the observed indicators, this study had to ensure that no pair of components exhibited a high correlation. Tables A1 and A2 illustrated correlations between the input indicators of personal car and truck models, respectively.

According to Mukaka [28], correlations between relevant variables should be less than 0.800, and the findings confirm that the statistical values fall within an acceptable range.

### 2.2. Decision Tree (DT)

For data analysis, this study used the DT model, which comprises two components [29]. The first component was the decision model structure, which comprises (a) the decision node, which functions as a node representing variables used for data sorting; (b) the branches representing the variables' values used to sort the data of each decision node; and (c) the leaf node showing the final result of sorting data of that variable. The second component was algorithms, which include (a) splitting, for selecting and dividing variable values in data sorting; (b) stopping, for controlling the model's establishment and termination based on specified conditions, ensuring that the model is not overfitting or underfitting; and (c) pruning, for adapting the model to optimize the model's suitability. This study applied the CART algorithm, which has the following advantages: (1) it analyzes both category and continuous variables [29]; (2) it has a binary splitting node format, suitably used for interpretation in crash data analysis [14]; (3) it analyzes the influence of the independent variables on the dependent variables [29], and uses the widely employed Gini algorithms.

### 2.3. Random Parameters (Mixed) Binary Logit Model

In this research, the mixed binary logit model was used to examine the factors affecting the severity of driver injuries as classified by the following involved vehicles: cars and trucks. This study adopted the random parameters binary logit model for the model analysis. The model begins by defining the severity function $S_{jm}$ of crash m sustaining injury severity $j$ as follows (Equation (1)) [12]:

$$S_{jm} = \beta_j X_{jm} + \varepsilon_{jm} \tag{1}$$

where $X_{jm}$ denotes a vector of the crash-level factors (independent variables) with $\beta_j$ as a vector of estimable parameters, and $\varepsilon_{jm}$ is an error term. Taking into account crash-specific unobserved heterogeneity, the outcome probabilities of random parameters logit model of car and truck driver-injury severities can be defined [30]

$$P_m(j) = \int \frac{\text{EXP}(\beta_j X_{jm})}{\sum_{\forall j} \text{EXP}(\beta_j X_{jm})} f(\beta|\rho) d\beta_j \tag{2}$$

where $P_m(j)$ defines the probability of driver injury severities $j$ in crash m, $f(\beta|\rho)$ is the density function of $\beta$ with $\rho$ being vector of parameters (means and variances). To account for possibility of unobserved heterogeneity in the means and variances of random parameters, the Equation is as follows

$$\beta_{jm} = \beta_j + \Phi_{jm} Z_{jm} + \sigma_{jm} \text{EXP}(\omega_{jm} W_{jm}) V_{jm} \tag{3}$$

where $\beta_{jm}$ is a vector of estimated parameters that varies across crashes. $\beta_j$ refers to the mean parameter estimate across all crashes, $Z_{jm}$ is a vector of the explanatory variable that captures heterogeneity in the mean that influences severities level $j$, $\Phi_{jm}$ represents a vector of estimable parameters, $W_{jm}$ refers to a vector of crashes-specific variables that captures heterogeneity in the standard deviation $\sigma_{jm}$ with corresponding vector $\omega_{jm}$, and disturbance term is denoted by $V_{jm}$.

### 2.4. Classification Accuracy

Model efficiency performance can be verified by using statistical values: true positive true negative, false positive, and false negative (Table 2). To build the metrics validating data of the model and covering model performance evaluation, we calculated the model by

using Equations (4)–(6), respectively. The obtained results from the model test [31] are as follows:

$$Accuracy = \frac{TP + TN}{TP + TN + FN + FP} \tag{4}$$

$$Sensitivity = \frac{TP}{TP + FN} \tag{5}$$

$$Specificity = \frac{TN}{FP + TN} \tag{6}$$

**Table 2.** Statistical values.

| | Predicted Positive (Fatal) | Predicted Negative (Non-Fatal) |
|---|---|---|
| Actual positive (fatal) | True Positive (*TP*) | False Negative (*FN*) |
| Actual negative (non-fatal) | False Positive (*FP*) | True Negative (*TN*) |

## 3. Results and Discussion

In this section, two statistical models (DT and RPBLHMV) were analyzed, and their independent variable importance was presented. Then, each model was used to test the most efficient model's performance in analyzing factors affecting the severity of run-off-road crashes (car and truck).

### 3.1. The Comparison of Model Prediction Accuracy

This study has compared the model accuracy for factors affecting drivers' injury severity in run-off-road crashes among cars and trucks on Thai highways. The presentation of DT and RPBLHMV found that each model could have advantages and disadvantages in the present study. The model's performance was measured based on its ability to predict with overall accuracy, sensitivity (predicting the true positive), and specificity (predicting the true negative), as shown in Table 3 below. We considered evaluating the ability to predict as a measure of the model's accuracy.

**Table 3.** Outcome prediction of decision tree and random parameters logit model.

| | | | Predicted | | | |
|---|---|---|---|---|---|---|
| | | | **Fatal (Car)** | **Non-Fatal (Car)** | **Fatal (Truck)** | **Non-Fatal (Truck)** |
| Actual | Decision tree | Fatal | 242 | 732 | 0 | 348 |
| | | Non-fatal | 190 | 2284 | 0 | 1027 |
| | Mixed logit | Fatal | 446 | 528 | 153 | 195 |
| | | Non-fatal | 159 | 2315 | 6 | 1021 |

In terms of the DT model (Table 4), the overall prediction accuracy between the car and truck models is quite similar. However, upon closer examination of sensitivity, it was observed that the truck model failed to predict any fatal crashes (0%). This limitation may stem from the smaller sample size of truck crashes (1375 cases), which falls below the required number for accurate measurement indicators. In light of previous findings by McNamara, et al. [32] and Genç and Mendeş [33], the necessary sample size may vary based on the type of measurement indicators or data used. Nevertheless, larger samples tend to yield more stable estimations and higher prediction accuracy.

While the car model exhibited a sensitivity greater than zero (24.85%), it still falls short when compared to the results obtained from mixed logit models. Notably, the prediction efficiency of the mixed logit models, encompassing both personal cars and trucks, proved intriguing. These models demonstrated the ability to correctly predict overall accuracy at a rate exceeding 80%, with sensitivity surpassing 40% and an almost perfect specificity

of close to 100%. The performance of the RPBLHMV model also aligns with prior literature [34], which has confirmed its superior predictive accuracy compared to traditional models due to its capability to capture hidden effects of unobserved heterogeneity (i.e., layer-2 effect) among crashes. Based on these findings, it can be concluded that the RPBLHMV models exhibit superiority over the DT models, particularly in terms of predicting sensitivity.

**Table 4.** Comparison of models' accuracy, sensitivity, and specificity.

| Method | Classification | Accuracy | Sensitivity | Specificity |
|---|---|---|---|---|
| Decision tree | Car | 73.26% | 24.85% | 92.32% |
| | Truck | 74.76% | 0% | 100% |
| Mixed logit | Car | 80.08% | 45.8% | 93.6% |
| | Truck | 85.38% | 43.97% | 99.42% |

Additionally, to effectively choose a model, a model with 0% accuracy for some factors can be a problem for model analysis when you need to predict some level of outcome (only non-fatal or fatal), not the overall outcome. Thus, a model with low prediction error must be selected for analyzing factors related to crashes [35,36]. The results of RPBLHMV, therefore, appeared to be an appropriate method for explaining and predicting the run-off-road crashes in this study, considering the low percentage of errors.

*3.2. Results of the Decision Tree Model*

3.2.1. Personal Car Classification

According to Figure 3, the results found four variables related to the dependent variable (injury severity). Off-road crashes on straight roads are the most variable factor that is significantly associated with the severity of a car driver's injury in a run-off-road crash. The results revealed that 53.9% of drivers who ran off the road on a straight route were more likely to die. There is evidence that straight roads (no curves) cause drivers to drive faster, resulting in greater injury severity when crashes occur (consistent with the finding of Obaid, et al. [37]). Further, the significant variable related to driver injury was a raised median [17]; the statistical results revealed that 36.5% of off-road drivers with a raised median have a greater chance of becoming more severe. Going off-road on curves is a significant variable in driver fatalities; the results revealed that off-road on curves cause drivers to fall into severe injury (36.5%) when compared to others [38]. This evidence could imply that run-off-road crashes on highways were found to be severe problems that had to be mitigated. Furthermore, 42.9% of drivers who suffer off-road on curves were found to be more likely to die when driving on dry roads, which is consistent with the findings of Peng and Boyle [39].

3.2.2. Truck Classification

The results of DT for truck drivers also found four variables related to the crash injury severity as well (as shown in Figure 4). Off-road crashes that go straight and strike safety equipment are the most significantly associated factor; results show that safety equipment (a safety barrier or guardrail) can save a driver's life (81.5% of crashes that strike safety equipment result in minor injury or PDO). Followed by mounted the traffic island, the results illustrated that truck drivers who are not mounting the island and have not off-straight road ben associated with minor injuries. Further, hitting safety equipment on curves is related to the level of severity (27.7% chance of fatal injuries). These findings suggest that a safety barrier or guardrail could reduce the severity of single-vehicle truck crashes (fewer fatalities) [40]. The last variable is road surface; this result is in line with the car model. The status of the road surface is one of the major factors that could influence the control of the vehicle while driving and result in levels of injury severity when suffering crashes, as confirmed by the evidence in related literature [39].

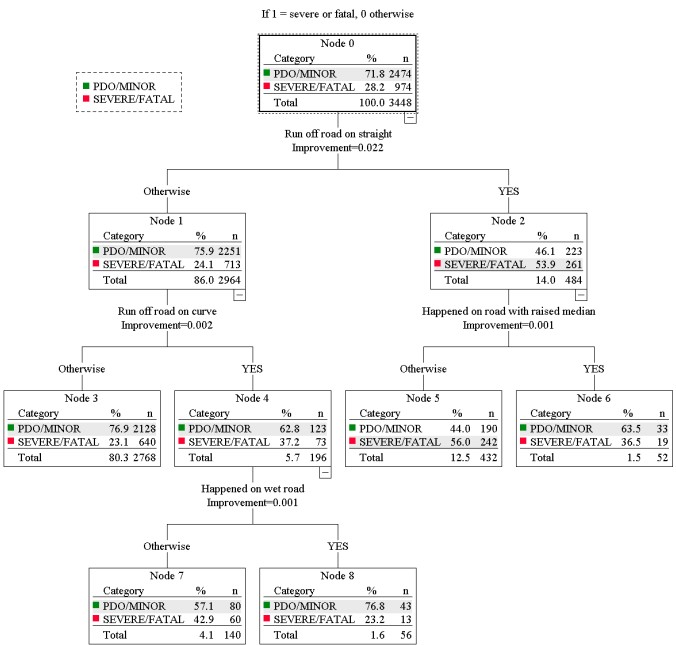

**Figure 3.** Model classification of car.

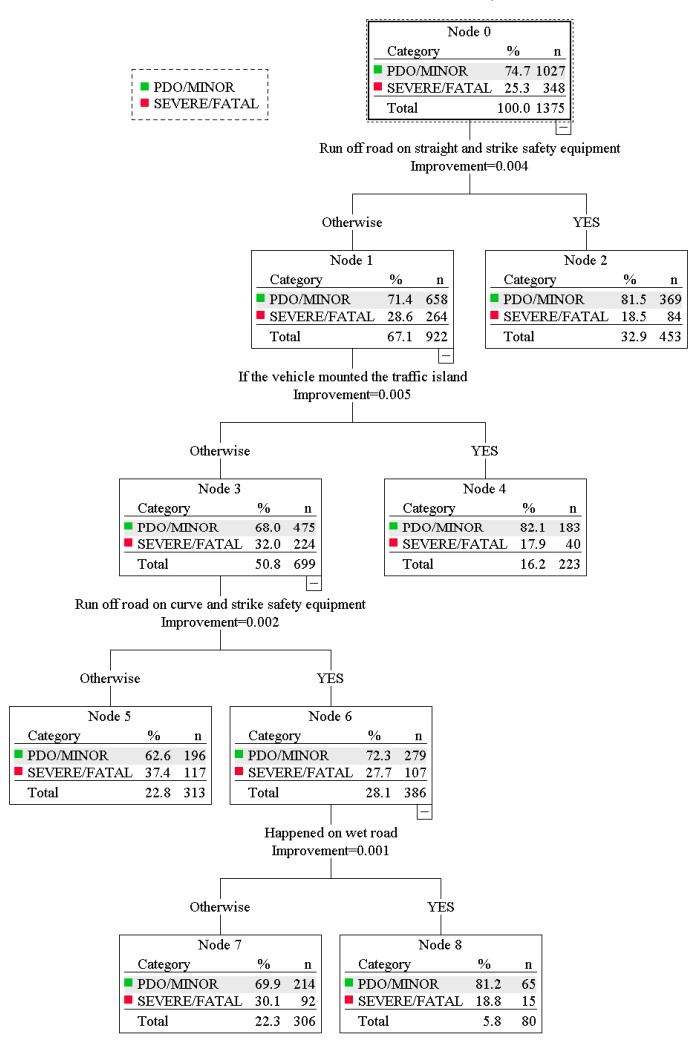

**Figure 4.** Model classification of truck.

### 3.3. Results of Random Parameters Binary Logit Model with Unobserved Heterogeneity

3.3.1. Explaining the Fix Parameters

According to Table 5 results, it was found that various factors related to run-off-road crashes could influence personal car and truck driver injury severity. The results found that McFadden $R^2$ of the car model was 0.0595 and 0.0502 for the truck model; that is, the variance of the data was explained at 5.95% and 5.02%, respectively [41]. The value of McFadden R2 is dependent on the data used, as confirmed by related literature [42–44], which indicated their McFadden $R^2$ ranges from 0.03 to 0.08. Although the model exhibits a low $R^2$, it still has potential in terms of explaining variables. Furthermore, the model's performance can be assessed through prediction ability, such as overall accuracy, sensitivity, and specificity, as shown in Table 4. Additionally, the RPBLHMV captures the random parameters of the model, which can impact the variance and introduce complexity for both car and truck drivers. Moreover, we tested the statistical fit of the model using a likelihood ratio test, as described in Equation (7):

$$\chi^2 = -2[LL(\beta_{RPBLHMV}) - LL(\beta_{withoutRP})] \tag{7}$$

**Table 5.** Model results of RPBLHMV (car and truck).

| Variables | CAR (*n* = 3448) | | | | TRUCK (*n* = 1375) | | | |
|---|---|---|---|---|---|---|---|---|
| | Estimates | S.E. | *t*-Stat | Marginal Effect | Estimates | S.E. | *t*-Stat | Marginal Effect |
| Constant | −0.421 | 0.333 | −1.26 | | 1.312 | 0.830 | 1.58 | |
| Non-random parameters; | | | | | | | | |
| AGE_26_35 | −0.04 | 0.082 | −0.49 | −0.009 | −0.194 | 0.19 | −1.02 | −0.042 |
| AGE_36_45 | 0.029 | 0.09 | 0.32 | 0.006 | 0.147 | 0.189 | 0.78 | 0.029 |
| AGE_46_55 | 0.027 | 0.106 | 0.26 | 0.006 | −0.089 | 0.211 | −0.42 | −0.02 |
| AGE_56_UP | 0.121 | 0.119 | 1.02 | 0.026 | 0.063 | 0.271 | 0.23 | 0.01 |
| MALE | 0.136 * | 0.073 | 1.87 | 0.029 | −0.156 | 0.503 | −0.31 | −0.032 |
| SAF_EQ | −0.162 *** | 0.061 | −2.64 | −0.034 | −0.085 | 0.114 | −0.74 | −0.018 |
| ALCOHOL | 0.263 | 0.206 | 1.28 | 0.056 | 0.612 | 0.587 | 1.04 | 0.126 |
| EXEED_SPEED | −0.089 | 0.085 | −1.05 | −0.019 | −0.147 | 0.141 | −1.04 | −0.03 |
| CONSTRUCT | 0.24 | 0.164 | 1.47 | 0.051 | −0.077 | 0.352 | −0.22 | −0.018 |
| ASPHALT | −0.021 | 0.106 | −0.2 | −0.005 | −0.299 | 0.192 | −1.56 | −0.061 |
| VERTICAL | | | | | 0.035 | 0.171 | 0.21 | 0.006 |
| INTERSECTION | | | | | −0.387 * | 0.217 | −1.78 | −0.081 |
| U_TURN | 0.012 | 0.115 | 0.11 | 0.003 | −0.206 | 0.254 | −0.81 | −0.043 |
| COMMUNITY | 0.427 | 0.269 | 1.59 | 0.091 | −0.622 | 0.758 | −0.82 | −0.131 |
| NO_MEDIAN | 0.025 | 0.278 | 0.09 | 0.005 | | | | |
| PAINTED | −0.19 | 0.306 | −0.62 | −0.04 | −0.874 | 0.666 | −1.31 | −0.181 |
| RAISED | | | | | −1.160 * | 0.614 | −1.89 | −0.24 |
| DEPRESSED | 0.106 | 0.277 | 0.38 | 0.023 | −0.907 | 0.609 | −1.49 | −0.188 |
| BARRIER | −0.043 | 0.301 | −0.14 | −0.009 | −1.090 * | 0.627 | −1.74 | −0.226 |
| MOUNT_ISLAND | −0.457 *** | 0.156 | −2.93 | −0.097 | −0.630 *** | 0.189 | −3.34 | −0.131 |
| PASS_IN_FRONT | −0.469 * | 0.257 | −1.82 | −0.1 | −0.142 | 0.319 | −0.45 | −0.028 |
| DEFECT_CAR | −0.661 * | 0.369 | −1.79 | −0.141 | −0.343 | 0.264 | −1.3 | −0.074 |
| WET_SURFACE | 0.099 | 0.205 | 0.48 | 0.021 | −0.11 | 0.329 | −0.33 | −0.022 |
| DIRTY_SURFACE | 0.456 | 0.463 | 0.98 | 0.097 | −0.966 | 1.567 | −0.62 | −0.21 |
| WEATHER | −0.243 | 0.198 | −1.23 | −0.052 | −0.217 | 0.327 | −0.66 | −0.045 |
| NIGHT | 0.079 | 0.062 | 1.27 | 0.017 | | | | |
| OFF_STR | 0.562 *** | 0.156 | 3.6 | 0.12 | | | | |
| OFF_STR_HIT | −0.458 *** | 0.151 | −3.03 | −0.097 | −0.698 *** | 0.156 | −4.48 | −0.144 |
| OFF_CUR | 0.125 | 0.183 | 0.68 | 0.027 | −0.023 | 0.242 | −0.09 | −0.006 |
| OFF_CUR_HIT | −0.538 *** | 0.162 | −3.33 | −0.115 | −0.440 ** | 0.177 | −2.48 | −0.092 |
| Random parameters; | | | | | | | | |
| VERTICAL | −0.194 | 0.133 | −1.46 | −0.041 | | | | |
| Standard deviation | 0.864 *** | 0.169 | 5.12 | | | | | |
| RAISED | −0.397 | 0.288 | −1.38 | −0.084 | | | | |
| Standard deviation | 1.963 ** | 0.145 | 13.55 | | | | | |
| NO_MEDIAN | | | | | −1.702 *** | 0.623 | −2.73 | −0.351 |
| Standard deviation | | | | | 4.161 *** | 0.475 | 8.76 | |

**Table 5.** *Cont.*

| Variables | CAR (n = 3448) | | | | TRUCK (n = 1375) | | | |
|---|---|---|---|---|---|---|---|---|
| | Estimates | S.E. | *t*-Stat | Marginal Effect | Estimates | S.E. | *t*-Stat | Marginal Effect |
| Heterogeneity in means; | | | | | | | | |
| VERTICAL: FALL_ASLEEP | 0.659 | 0.419 | 1.57 | | | | | |
| RAISED: FALL_ASLEEP | −0.470 * | 0.268 | −1.75 | | | | | |
| NO_MEDIAN: FALL_ASLEEP | | | | | 1.364 *** | 0.476 | −2.86 | |
| Heterogeneity in variance; | | | | | | | | |
| VERTICAL: INTERSECTION | −0.346 | 1.439 | −0.24 | | | | | |
| RAISED: INTERSECTION | −3.326 *** | 0.206 | −16.13 | | | | | |
| NO_MEDIAN: NIGHT | | | | | −0.244 * | 0.143 | −1.71 | |
| NO_MEDIAN: OFF_STR | | | | | 3.998 *** | 1.261 | 3.17 | |

Model statistics: S.E. = standard error; Halton draw = 1000; $AIC_{car}$ = 3932.7; $AIC_{truck}$ = 1544.7; ***, **, * Significance at 1%, 5%, 10% level, respectively.

The results demonstrate that the RPBLHMV significantly outperforms the traditional model (without random parameters and heterogeneities). Specifically, we observed a significant improvement in the RPBLHMV for both car and truck models at a 99% confidence interval.

The RPBLHMV model was analyzed in terms of factors related to non-fatal or fatal crashes. The driver factors of personal cars with seat-belt-wearing behavior were significantly related to the reduction in deaths from crashes because wearing seat belts helps prevent the driver's physical severity in injury from being crushed in the crash and bouncing off the vehicle, which is consistent with related literature [12]. Following gender, the findings revealed that male drivers were more likely to sustain serious injuries in run-off-road collisions; this finding is in line with Al-Balbissi [45], who reported that there was a definite trend toward significantly higher accident rates for male drivers compared with female drivers.

Factors related to crash characteristics have a significant effect on the risk faced by private car and truck occupants. A finding of this study was that road crashes within a vehicle mounted on the median had fewer chances of death for private car and truck users [12]. Furthermore, personal car crashes that are caused by passing in front of an occupant car and defective equipment of vehicle influence the drivers to become less likely to die. This is consistent with a finding of Behnood and Mannering [46]. Additionally, the truck model illustrated that a road divided by raised or barrier median could potentially save truck drivers from the risk of fatality; this finding is in accordance with relevant literature [11,46].

Further, personal car and truck drivers who have a crash by driving vehicles off the road, whether on the straight or curve, and hitting safety equipment (barrier or guardrail) on the roadside, the likelihood of fatal crashes is potentially reduced (the same is true for truck crashes with a barrier median). This finding is consistent with that of Roque, et al. [40] and Chitturi, et al. [47]; that is, the unavailability of roadside safety equipment in the case of run-off-road crashes would result in increasing fatality. Roque, et al. [40] also revealed that roadside features such as safety barriers and guardrails significantly reduce the fatality risk for drivers. This result is logical and meets the purpose of the implementation. In addition, this study also found that roadsides without safety barriers or guardrails could cause severe injuries to car and truck drivers.

### 3.3.2. Influence of Random Parameters and Unobserved Heterogeneity

In the case of random parameters, the factors that have the potential to be random parameters of car drivers are raised, and barrier median. This study found that car drivers who have a crash at a raised median or road with a slope are less likely to die. These results also found truck drivers who have crashed at the no-road divider areas tend to decrease the level of severity.

This study also captured the unobserved heterogeneity in the means of the data. The personal car results illustrated that crashes caused by falling asleep can increase the likelihood of death in crashes on grading roads [48]. In contrast, fall-asleep indicators have been found to decrease injury severity when a crash occurred on the raised median. For truck drivers, falling asleep also influences the injury severity of crashes at no median road. In addition, The result of the personal car model illustrated the crash occurred at an intersection as representative of heterogeneity in variances that can decrease the variation of injury severity of crash at raised median and slope. A previous study reported that intersection area generally creates a number of conflicts with traffic [49]; as a result, they become a cautious area where driver drive with greater caution and slow their vehicle down while driving within these areas. Therefore, even if encountering a crash, it probably does not cause more serious injury to the driver (this is consistent with Ma, et al. [50]). Regarding truck results, driving at nighttime and off-road crash on strength could influence the proportion of death in truck driver crashes.

## 4. Conclusions

The purpose of this study was to compare two potential analysis concepts (data-driven and econometric analysis) in the study of run-off-road injury severity. The data was obtained from Thailand's Department of Highways, which contained statistics on car and truck run-off-road crashes on highways between 2011 and 2017. The dependent variable is divided into two categories consisting of non-fatal and fatal injuries. The study results are presented as follows:

Regarding DT results, it was indicated that there is a difference between the variables important to the model among personal car and truck run-off-road crashes on highways. Off-road conditions on straight and curved roads raised medians, and wet surfaces were found to be of variable importance in causing car crashes in this study. Furthermore, wet surfaces, off-roading with striking safety equipment (both straight and curved), and traffic island mounting were discovered to be of variable importance depending on the truck model.

According to the RPBLHMV analysis results, factors associated with crash severity were classified as non-random, random parameters, and unobserved heterogeneity in means and variances. The car model demonstrates that the driver's use of a seat belt reduces the risk of fatality. Crash characteristics, including off-roading with striking safety equipment, can reduce the risk of death. Furthermore, raised median and slope were representations of the model random parameter. Falling asleep and encountering intersections play a heterogeneous role in means and variances, respectively. For the truck model, it was found that intersections, raised and barrier medians, and mounted traffic islands can affect the crash injury severity, and it was also captured that crashes at no median area could influence the model's variation. In addition, falling asleep, nighttime, and strong off-road crash represented the model's heterogeneity in means and variance.

Practical implications arise from considering the associated factors, thereby emphasizing the need for relevant agencies involved in policy design to prioritize certain measures. These include promoting legislation on seat belt usage [51] and enhancing knowledge through safe driving training courses, enabling drivers to exercise caution and attentiveness while operating vehicles [52]. These factors hold a significant influence over personal car drivers. Additionally, the potential of guardrails and other safety equipment in reducing fatality risks for both car and truck drivers, as observed in both RPBLHMV and DT models, suggests the importance of strategically installing such safety measures to enhance overall road safety effectiveness.

Furthermore, the RPBLHMV model demonstrates its ability to capture unobserved heterogeneity, which plays a crucial role in accounting for hidden effects and enhancing the explanatory power of the model. On the other hand, the DT model offers a straightforward approach to identifying variable importance in relation to injury severity by prioritizing

factors through sequencing, thereby providing valuable insights into the most significant factors.

In terms of the comparative method employed in this study, it reveals the advantages and disadvantages of utilizing both machine learning (ML) and econometric analysis concepts. This comparison enables authorities to make informed model choices that align with their specific objectives and facilitates the design of appropriate measures and policies accordingly. By understanding the strengths and weaknesses of each approach, decision-makers can effectively tailor their strategies to achieve desired outcomes.

In terms of related areas, as per recommendations, around the routes prone to crashes, the involved agencies should design protective equipment in such areas. In addition, to reduce the likelihood of death and the damage associated with other road users from run-off-road collisions, safety barriers and guard rails should be installed to prevent vehicles from deviating from the routes [53].

As per research limitations, this study only focused on factors associated with run-off-road crashes; this type of crash may have some different attributes when compared to others [54]; a study on another type of crash could produce different results. In this study, DT can be easily used to provide variable importance in the model but has limitations in terms of providing direction for independent variables on crash severity. However, the RPBLHMV has the potential to address such a problem and showed greater overall prediction accuracy than DT. However, the McFadden R2 and AIC scores remain relatively low, particularly for the truck model. This indicates that despite the overall improvement in model accuracy compared to the DT approach, there is still a limited extent to which the model explains the variance in the data. To enhance the model's explanatory power in future research, it may be necessary to adjust the model parameters or consider alternative methods that better fit the data. Furthermore, it is recommended to explore comparative methods utilized in other geographical areas beyond Thailand, as well as conduct more comprehensive comparisons. Incorporating numerical values from relevant literature into the comparative analysis can yield more efficient and effective results. By broadening the scope of the comparison and delving deeper into the existing body of knowledge, researchers can gain valuable insights and enhance the robustness of their findings. Based on the specific advantages and disadvantages of data-driven and econometric analysis, the method potentially empowers researchers to determine the method appropriate to the educational context.

**Author Contributions:** Conceptualization, T.C. and P.W.; methodology, T.C. and C.B.; software, S.J. and V.R.; validation, C.S.; formal analysis, P.W.; resources, V.R.; data curation, T.C.; writing—original draft preparation, T.C. and P.W.; writing—review and editing, P.W.; visualization, C.S.; supervision, C.B.; project administration, V.R.; funding acquisition, S.J. All authors have read and agreed to the published version of the manuscript.

**Funding:** This research was supported by (i) Suranaree University of Technology (SUT), (ii) Thailand Science Research and Innovation (TSRI), and (iii) the National Science, Research and Innovation Fund (NSRF) (project code: 4284945) (Grant number: Full-time61/02/2566).

**Institutional Review Board Statement:** The study was conducted according to the guidelines of the Declaration of Helsinki and approved by the Ethics Committee of Suranaree University of Technology (COE.5/2565).

**Informed Consent Statement:** Not applicable.

**Data Availability Statement:** Data available on request due to restrictions.

**Conflicts of Interest:** The authors declare no conflict of interest.

# Appendix A

**Table A1.** Correlations between related indicators of personal car data.

| | SEV | V1 | V2 | V3 | V4 | V5 | V6 | V7 | V8 | V9 | V10 | V11 | V12 | V13 | V14 | V15 |
|---|---|---|---|---|---|---|---|---|---|---|---|---|---|---|---|---|
| SEV | 1 | −0.019 | 0.006 | −0.002 | 0.022 | 0.031 | −0.034 * | 0.040 * | −0.028 | 0.035 * | 0.036 * | −0.009 | −0.028 | −0.022 | −0.024 | 0.020 |
| V1 | −0.019 | 1 | −0.403 ** | −0.288 ** | −0.229 ** | −0.041 * | −0.008 | 0.033 | −0.002 | −0.015 | −0.021 | 0.055 ** | 0.014 | −0.021 | −0.040 * | 0.032 |
| V2 | 0.006 | −0.403 ** | 1 | −0.202 ** | −0.161 ** | 0.022 | 0.014 | −0.012 | −0.005 | −0.004 | 0.018 | −0.012 | 0.002 | 0.011 | 0.051 ** | −0.019 |
| V3 | −0.002 | −0.288 ** | −0.202 ** | 1 | −0.115 ** | 0.006 | 0.002 | −0.024 | −0.010 | 0.011 | −0.007 | 0.007 | −0.013 | −0.003 | 0.012 | 0.005 |
| V4 | 0.022 | −0.229 ** | −0.161 ** | −0.115 ** | 1 | 0.069 ** | 0.031 | −0.024 | −0.018 | 0.034 * | 0.005 | −0.024 | 0.012 | 0.014 | 0.009 | 0.001 |
| V5 | 0.031 | −0.041 * | 0.022 | 0.006 | 0.069 ** | 1 | −0.016 | 0.003 | 0.016 | 0.001 | −0.020 | −0.029 | 0.007 | 0.015 | 0.021 | 0.020 |
| V6 | −0.034 * | −0.008 | 0.014 | 0.002 | 0.031 | −0.016 | 1 | 0.029 | −0.061 ** | 0.050 ** | 0.017 | 0.074 ** | 0.039 * | 0.012 | 0.004 | 0.004 |
| V7 | 0.040 * | 0.033 | −0.012 | −0.024 | −0.024 | 0.003 | 0.029 | 1 | −0.132 ** | −0.029 | 0.004 | 0.010 | 0.039 * | 0.024 | −0.022 | 0.031 |
| V8 | −0.028 | −0.002 | −0.005 | −0.010 | −0.018 | 0.016 | −0.061 ** | −0.132 ** | 1 | −0.756 ** | −0.005 | −0.003 | 0.044 * | −0.047 ** | 0.018 | −0.009 |
| V9 | 0.035 * | −0.015 | −0.004 | 0.011 | 0.034 * | 0.001 | 0.050 ** | −0.029 | −0.756 ** | 1 | −0.004 | 0.010 | −0.055 ** | 0.020 | −0.011 | −0.011 |
| V10 | 0.036 * | −0.021 | 0.018 | −0.007 | 0.005 | −0.020 | 0.017 | 0.004 | −0.005 | −0.004 | 1 | −0.039 * | −0.009 | 0.055 ** | 0.020 | 0.000 |
| V11 | −0.009 | 0.055 ** | −0.012 | 0.007 | −0.024 | −0.029 | 0.074 ** | 0.010 | −0.003 | 0.010 | −0.039 * | 1 | 0.059 ** | −0.030 | −0.034 * | 0.011 |
| V12 | −0.028 | 0.014 | 0.002 | −0.013 | 0.012 | 0.007 | 0.039 * | 0.039 * | 0.044 * | −0.055 ** | −0.009 | 0.059 ** | 1 | −0.044 ** | −0.066 ** | −0.021 |
| V13 | −0.022 | −0.021 | 0.011 | −0.003 | 0.014 | 0.015 | 0.012 | 0.024 | -.047 ** | 0.020 | 0.055 ** | −0.030 | −0.044 ** | 1 | 0.106 ** | 0.107 ** |
| V14 | −0.024 | −0.040 * | 0.051 ** | 0.012 | 0.009 | 0.021 | 0.004 | −0.022 | 0.018 | −0.011 | 0.020 | −0.034 * | −0.066 ** | 0.106 ** | 1 | 0.015 |
| V15 | 0.020 | 0.032 | −0.019 | 0.005 | 0.001 | 0.020 | 0.004 | 0.031 | −0.009 | −0.011 | 0.000 | 0.011 | −0.021 | 0.107 ** | 0.015 | 1 |
| V16 | 0.021 | 0.011 | −0.026 | 0.029 | 0.014 | 0.021 | −0.010 | 0.050 ** | −0.074 ** | 0.026 | 0.007 | 0.125 ** | 0.127 ** | −0.039 * | −0.187 ** | −0.002 |
| V17 | −0.034 * | 0.041 * | −0.003 | −0.048 ** | −0.008 | 0.022 | 0.024 | −0.021 | −0.064 ** | 0.036 * | 0.008 | 0.049 ** | 0.019 | 0.018 | −0.063 ** | 0.003 |
| V18 | −0.048 ** | −0.003 | −0.016 | −0.017 | 0.007 | 0.008 | −0.049 ** | 0.006 | 0.064 ** | −0.070 ** | −0.005 | −0.084 ** | −0.066 ** | 0.115 ** | 0.279 ** | 0.034 * |
| V19 | 0.042 * | −0.030 | 0.028 | 0.013 | 0.010 | −0.020 | 0.066 ** | −0.036 * | 0.028 | 0.037 * | −0.029 | −0.030 | −0.098 ** | −0.088 ** | −0.037 * | −0.025 |
| V20 | −0.004 | 0.018 | 0.020 | −0.002 | −0.055 ** | −0.036 * | −0.038 * | −0.020 | 0.023 | −0.026 | 0.057 ** | −0.070 ** | 0.059 ** | −0.006 | −0.040 * | −0.010 |
| V21 | −0.055 ** | −0.038 * | 0.030 | −0.004 | 0.001 | 0.004 | −0.058 ** | −0.015 | 0.062 ** | −0.046 ** | −0.017 | −0.063 ** | −0.140 ** | 0.101 ** | 0.330 ** | 0.015 |
| V22 | −0.024 | −0.016 | 0.000 | 0.028 | −0.005 | 0.012 | 0.013 | −0.019 | −0.291 ** | −0.052 ** | 0.001 | −0.022 | −0.006 | 0.034 * | 0.016 | 0.027 |
| V23 | −0.023 | −0.013 | 0.030 | 0.029 | −0.015 | −0.003 | 0.015 | −0.015 | −0.228 ** | −0.041 * | −0.003 | −0.031 | −0.025 | 0.000 | 0.008 | −0.011 |
| V24 | −0.030 | −0.020 | 0.086 ** | 0.001 | −0.008 | −0.020 | 0.047 ** | 0.010 | 0.121 ** | −0.111 ** | −0.015 | 0.063 ** | 0.104 ** | −0.012 | −0.007 | −0.027 |
| V25 | 0.014 | −0.007 | 0.013 | −0.023 | 0.016 | −0.001 | −0.003 | −0.008 | 0.018 | −0.023 | 0.046 ** | 0.019 | −0.019 | 0.020 | 0.043 * | −0.006 |
| V26 | −0.036 * | 0.000 | 0.068 ** | −0.002 | −0.006 | −0.017 | 0.031 | 0.017 | 0.124 ** | −0.116 ** | −0.017 | 0.062 ** | 0.085 ** | −0.004 | −0.002 | −0.023 |
| V27 | 0.001 | 0.068 ** | −0.019 | −0.050 ** | −0.112 ** | 0.078 ** | −0.025 | 0.079 ** | −0.002 | 0.014 | 0.016 | −0.051 ** | −0.067 ** | 0.064 ** | 0.099 ** | 0.008 |
| V28 | 0.230 ** | −0.003 | 0.000 | −0.003 | 0.016 | −0.018 | 0.015 | 0.042 * | −0.092 ** | 0.096 ** | 0.041 * | 0.026 | −0.085 ** | −0.066 ** | −0.114 ** | −0.024 |
| V29 | −0.084 ** | −0.005 | −0.020 | −0.010 | −0.001 | 0.006 | −0.024 | −0.040 * | −0.045 ** | 0.059 ** | −0.010 | −0.004 | −0.171 ** | 0.015 | −0.087 ** | 0.017 |
| V30 | 0.049 ** | 0.017 | −0.004 | 0.012 | 0.007 | 0.019 | 0.030 | 0.006 | −0.001 | −0.040 * | 0.011 | 0.037 * | 0.153 ** | −0.024 | −0.073 ** | −0.012 |
| V31 | −0.085 ** | 0.047 ** | −0.014 | 0.011 | −0.022 | −0.006 | 0.057 ** | 0.009 | 0.072 ** | −0.065 ** | −0.034 * | 0.055 ** | 0.360 ** | −0.051 ** | −0.136 ** | −0.020 |

**Table A1.** *Cont.*

|  | V16 | V17 | V18 | V19 | V20 | V21 | V22 | V23 | V24 | V25 | V26 | V27 | V28 | V29 | V30 | V31 |
|---|---|---|---|---|---|---|---|---|---|---|---|---|---|---|---|---|
| V16 | 1 | −0.140 ** | −0.376 ** | −0.438 ** | −0.138 ** | −0.291 ** | 0.013 | −0.013 | −0.027 | 0.016 | −0.011 | 0.004 | 0.069 ** | 0.066 ** | 0.087 ** | 0.146 ** |
| V17 | −0.140 ** | 1 | −0.144 ** | −0.168 ** | −0.053 ** | −0.075 ** | 0.024 | 0.046 ** | −0.001 | −0.014 | −0.010 | 0.046 ** | −0.025 | 0.074 ** | −0.006 | 0.025 |
| V18 | −0.376 ** | −0.144 ** | 1 | −0.452 ** | −0.142 ** | 0.397 ** | 0.023 | 0.007 | 0.031 | 0.035 * | 0.035 * | 0.124 ** | −0.155 ** | −0.220 ** | −0.047 ** | −0.051 ** |
| V19 | −0.438 ** | −0.168 ** | −0.452 ** | 1 | −0.166 ** | −0.024 | −0.041 * | −0.014 | −0.018 | −0.035 * | −0.027 | −0.128 ** | 0.090 ** | 0.094 ** | −0.057 ** | −0.100 ** |
| V20 | −0.138 ** | −0.053 ** | −0.142 ** | −0.166 ** | 1 | −0.082 ** | −0.004 | −0.001 | 0.031 | −0.014 | 0.021 | −0.027 | 0.000 | 0.048 ** | 0.042 * | −0.013 |
| V21 | −0.291 ** | −0.075 ** | 0.397 ** | −0.024 | −0.082 ** | 1 | −0.009 | −0.021 | −0.003 | 0.041 * | 0.017 | 0.135 ** | −0.232 ** | −0.397 ** | −0.141 ** | −0.279 ** |
| V22 | 0.013 | 0.024 | 0.023 | −0.041 * | −0.004 | −0.009 | 1 | −0.016 | −0.044 * | −0.009 | −0.053 ** | −0.038 * | −0.009 | 0.040 * | 0.001 | −0.042 * |
| V23 | −0.013 | 0.046 ** | 0.007 | −0.014 | −0.001 | −0.021 | −0.016 | 1 | −0.026 | −0.007 | −0.029 | −0.027 | 0.016 | 0.014 | 0.007 | −0.027 |
| V24 | −0.027 | −0.001 | 0.031 | −0.018 | 0.031 | −0.003 | −0.044 * | −0.026 | 1 | 0.026 | 0.888 ** | −0.088 ** | −0.033 | −0.094 ** | 0.089 ** | 0.091 ** |
| V25 | 0.016 | −0.014 | 0.035 * | −0.035* | −0.014 | 0.041 * | −0.009 | −0.007 | 0.026 | 1 | 0.023 | 0.033 | −0.011 | −0.032 | 0.005 | −0.006 |
| V26 | −0.011 | −0.010 | 0.035* | −0.027 | 0.021 | 0.017 | −0.053 ** | −0.029 | 0.888 ** | 0.023 | 1 | −0.044 * | −0.029 | −0.102 ** | 0.083 ** | 0.081 ** |
| V27 | 0.004 | 0.046 ** | 0.124 ** | −0.128 ** | −0.027 | 0.135 ** | −0.038 * | −0.027 | −0.088 ** | 0.033 | −0.044 * | 1 | −0.066 ** | −0.013 | −0.087 ** | −0.022 |
| V28 | 0.069 ** | −0.025 | −0.155 ** | 0.090 ** | 0.000 | −0.232 ** | −0.009 | 0.016 | −0.033 | −0.011 | −0.029 | −0.066 ** | 1 | −0.279 ** | −0.099 ** | −0.196 ** |
| V29 | 0.066 ** | 0.074 ** | −0.220 ** | 0.094 ** | 0.048 ** | −0.397 ** | 0.040 * | 0.014 | −0.094 ** | −0.032 | −0.102 ** | −0.013 | −0.279 ** | 1 | −0.170 ** | −0.336 ** |
| V30 | 0.087 ** | −0.006 | −0.047 ** | −0.057 ** | 0.042 * | −0.141 ** | 0.001 | 0.007 | 0.089 ** | 0.005 | 0.083 ** | −0.087 ** | −0.099 ** | −0.170 ** | 1 | −0.119 ** |
| V31 | 0.146 ** | 0.025 | −0.051 ** | −0.100 ** | −0.013 | −0.279 ** | −0.042 * | −0.027 | 0.091 ** | −0.006 | 0.081 ** | −0.022 | −0.196 ** | −0.336 ** | −0.119 ** | 1 |

Note: ** indicates that correlation is significant at 0.01 level (2-tailed). * indicates that correlation is significant at 0.05 level (2-tailed).

**Table A2.** Correlations between related indicators of truck data.

| | SEV | V1 | V2 | V3 | V4 | V5 | V6 | V7 | V8 | V9 | V10 | V11 | V12 | V13 | V14 | V15 |
|---|---|---|---|---|---|---|---|---|---|---|---|---|---|---|---|---|
| SEV | 1 | | | | | | | | | | | | | | | |
| V1 | −0.042 | 1 | | | | | | | | | | | | | | |
| V2 | 0.038 | −0.485 ** | 1 | | | | | | | | | | | | | |
| V3 | −0.004 | −0.336 ** | −0.326 ** | 1 | | | | | | | | | | | | |
| V4 | 0.020 | −0.176 ** | −0.171 ** | −0.118 ** | 1 | | | | | | | | | | | |
| V5 | 0.001 | −0.033 | 0.031 | 0.004 | 0.023 | 1 | | | | | | | | | | |
| V6 | −0.028 | 0.020 | 0.002 | −0.002 | −0.039 | 0.006 | 1 | | | | | | | | | |
| V7 | 0.009 | −0.024 | 0.015 | 0.026 | −0.021 | 0.008 | −0.011 | 1 | | | | | | | | |
| V8 | −0.036 | 0.047 | 0.009 | −0.045 | 0.012 | −0.010 | 0.003 | −0.038 | 1 | | | | | | | |
| V9 | 0.030 | 0.012 | −0.008 | 0.008 | −0.056 * | 0.015 | −0.034 | −0.010 | −0.622 ** | 1 | | | | | | |
| V10 | −0.001 | −0.010 | 0.014 | −0.007 | −0.002 | 0.015 | 0.028 | −0.014 | 0.017 | −0.027 | 1 | | | | | |
| V11 | −0.025 | −0.027 | 0.028 | −0.023 | 0.017 | −0.025 | 0.050 | −0.011 | −0.078 ** | 0.033 | −0.011 | 1 | | | | |
| V12 | 0.070 ** | −0.013 | 0.013 | 0.022 | 0.024 | 0.047 | 0.052 | −0.043 | 0.005 | −0.133 ** | −0.002 | 0.046 | 1 | | | |
| V13 | −0.070 ** | −0.016 | 0.020 | −0.004 | −0.009 | 0.001 | 0.009 | 0.005 | 0.038 | −0.063 * | 0.032 | −0.054 * | −0.106 ** | 1 | | |
| V14 | −0.054 * | −0.004 | 0.016 | −0.026 | 0.007 | −0.045 | 0.007 | 0.016 | 0.013 | −0.007 | 0.000 | 0.002 | −0.098 ** | 0.073 ** | 1 | |
| V15 | −0.019 | −0.017 | 0.003 | −0.024 | 0.010 | 0.009 | 0.043 | −0.008 | −0.008 | −0.038 | −0.015 | 0.025 | −0.027 | 0.055 * | 0.011 | 1 |
| V16 | 0.080 ** | −0.048 | 0.021 | 0.036 | 0.012 | 0.036 | 0.032 | 0.027 | −0.079 ** | −0.047 | −0.016 | 0.130 ** | 0.377 ** | −0.088 ** | −0.173 ** | −0.020 |
| V17 | −0.006 | 0.018 | −0.049 | 0.003 | 0.010 | −0.029 | −0.037 | −0.015 | 0.004 | −0.011 | 0.024 | −0.003 | −0.036 | 0.037 | −0.043 | 0.120 ** |
| V18 | −0.087 ** | 0.048 | −0.023 | −0.014 | −0.012 | −0.037 | −0.071 ** | 0.004 | 0.049 | −0.060 * | −0.018 | −0.146 ** | −0.164 ** | 0.202 ** | 0.165 ** | −0.044 |
| V19 | −0.004 | 0.009 | −0.009 | −0.019 | −0.007 | 0.022 | 0.076 ** | −0.011 | 0.004 | 0.126 ** | 0.009 | 0.008 | −0.248 ** | −0.078 ** | 0.079 ** | 0.027 |
| V20 | −0.020 | −0.022 | 0.041 | −0.004 | 0.006 | −0.036 | −0.068 * | −0.023 | 0.070 ** | −0.052 | 0.027 | −0.030 | 0.022 | −0.011 | −0.041 | −0.026 |
| V21 | −0.075 ** | 0.028 | −0.023 | −0.046 | 0.009 | −0.022 | −0.065 * | 0.032 | 0.057 * | −0.036 | −0.035 | 0.031 | −0.195 ** | 0.057 * | 0.279 ** | 0.022 |
| V22 | −0.013 | −0.026 | 0.005 | 0.029 | −0.028 | −0.027 | 0.048 | −0.016 | −0.283 ** | −0.074 ** | −0.030 | 0.049 | −0.061 * | 0.046 | 0.026 | 0.115 ** |
| V23 | 0.022 | −0.081 ** | −0.005 | 0.056 * | 0.072 ** | −0.007 | 0.015 | 0.012 | −0.407 ** | −0.107 ** | −0.007 | 0.024 | 0.111 ** | −0.017 | −0.026 | −0.025 |
| V24 | −0.064 * | −0.013 | 0.017 | 0.054 * | −0.023 | 0.008 | 0.007 | 0.043 | 0.189 ** | −0.133 ** | −0.013 | 0.068 * | −0.026 | 0.037 | 0.053 | 0.031 |
| V25 | −0.013 | 0.000 | −0.022 | −0.003 | 0.078 ** | 0.006 | 0.042 | −0.006 | −0.029 | 0.005 | 0.058 * | 0.018 | 0.022 | 0.019 | −0.016 | −0.006 |
| V26 | −0.070 ** | −0.006 | 0.020 | 0.055 * | −0.031 | 0.011 | 0.029 | 0.039 | 0.186 ** | −0.149 ** | −0.007 | 0.075 ** | −0.024 | 0.045 | 0.043 | 0.028 |
| V27 | 0.000 | 0.022 | 0.030 | −0.066 * | −0.021 | 0.000 | 0.087 ** | −0.003 | −0.058 * | 0.138 ** | 0.028 | −0.010 | −0.004 | −0.036 | 0.032 | −0.016 |
| V28 | 0.090 ** | 0.013 | −0.011 | −0.016 | −0.005 | 0.003 | −0.023 | 0.031 | −0.132 ** | 0.159 ** | −0.006 | 0.056* | −0.097 ** | −0.035 | −0.045 | −0.003 |
| V29 | −0.109 ** | 0.023 | 0.007 | −0.029 | −0.016 | −0.001 | 0.055 * | −0.005 | −0.016 | 0.064 * | 0.011 | −0.097 ** | −0.268 ** | 0.047 | −0.029 | 0.001 |
| V30 | 0.083 ** | −0.007 | −0.017 | 0.034 | 0.008 | 0.028 | −0.042 | −0.025 | −0.010 | −0.072 ** | 0.053 * | 0.046 | 0.244 ** | −0.070 ** | −0.071 ** | −0.028 |
| V31 | 0.035 | −0.029 | 0.005 | 0.050 | 0.018 | −0.011 | 0.055 * | −0.034 | 0.069 * | −0.108 ** | −0.021 | 0.038 | 0.373 ** | −0.046 | −0.110 ** | −0.024 |



**Table A2.** *Cont.*

| | V16 | V17 | V18 | V19 | V20 | V21 | V22 | V23 | V24 | V25 | V26 | V27 | V28 | V29 | V30 | V31 |
|---|---|---|---|---|---|---|---|---|---|---|---|---|---|---|---|---|
| V16 | 1 | −0.131 ** | −0.346 ** | −0.555 ** | −0.200 ** | −0.282 ** | −0.067 * | 0.113 ** | −0.085 ** | −0.002 | −0.056 * | −0.022 | −0.042 | −0.182 ** | 0.173 ** | 0.344 ** |
| V17 | −0.131 ** | 1 | −0.083 ** | −0.134 ** | −0.048 | −0.055 * | 0.039 | −0.030 | 0.007 | −0.012 | −0.010 | 0.013 | 0.001 | 0.100 ** | −0.036 | −0.045 |
| V18 | −0.346 ** | −0.083 ** | 1 | −0.355 ** | −0.128 ** | 0.254 ** | 0.020 | −0.010 | 0.096 ** | 0.055 * | 0.101 ** | −0.049 | −0.073 ** | −0.032 | −0.061 * | −0.117 ** |
| V19 | −0.555 ** | −0.134 ** | −0.355 ** | 1 | −0.206 ** | 0.127 ** | 0.014 | −0.084 ** | −0.002 | −0.027 | −0.023 | 0.068 * | 0.123 ** | 0.150 ** | −0.143 ** | −0.229 ** |
| V20 | −0.200 ** | −0.048 | −0.128 ** | −0.206 ** | 1 | −0.058 * | 0.030 | −0.026 | 0.015 | −0.018 | −0.002 | −0.027 | −0.038 | 0.047 | 0.058 * | −0.011 |
| V21 | −0.282 ** | −0.055 * | 0.254 ** | 0.127 ** | −0.058 * | 1 | 0.052 | −0.021 | 0.077 ** | 0.031 | 0.079 ** | 0.020 | −0.141 ** | −0.308 ** | −0.129 ** | −0.275 ** |
| V22 | −0.067 * | 0.039 | 0.020 | 0.014 | 0.030 | 0.052 | 1 | −0.049 | −0.041 | −0.012 | −0.036 | −0.031 | −0.003 | 0.028 | −0.039 | −0.069 * |
| V23 | 0.113 ** | −0.030 | −0.010 | −0.084 ** | −0.026 | −0.021 | −0.049 | 1 | −0.117 ** | −0.018 | −0.095 ** | −0.098 ** | −0.024 | −0.048 | 0.075 ** | 0.031 |
| V24 | −0.085 ** | 0.007 | 0.096 ** | −0.002 | 0.015 | 0.077 ** | −0.041 | −0.117 ** | 1 | −0.005 | 0.923 ** | −0.075 ** | −0.022 | −0.051 | 0.009 | 0.012 |
| V25 | −0.002 | −0.012 | 0.055 * | −0.027 | −0.018 | 0.031 | −0.012 | −0.018 | −0.005 | 1 | 0.019 | 0.011 | −0.021 | 0.001 | 0.021 | −0.017 |
| V26 | −0.056 * | −0.010 | 0.101 ** | −0.023 | −0.002 | 0.079 ** | −0.036 | −0.095 ** | 0.923 ** | 0.019 | 1 | −0.040 | −0.026 | −0.062 * | −0.008 | 0.039 |
| V27 | −0.022 | 0.013 | −0.049 | 0.068 * | −0.027 | 0.020 | −0.031 | −0.098 ** | −0.075 ** | 0.011 | −0.040 | 1 | 0.013 | 0.057 * | −0.106 ** | 0.020 |
| V28 | −0.042 | 0.001 | −0.073 ** | 0.123 ** | −0.038 | −0.141 ** | −0.003 | −0.024 | −0.022 | −0.021 | −0.026 | 0.013 | 1 | −0.225 ** | −0.094 ** | −0.200 ** |
| V29 | −0.182 ** | 0.100 ** | −0.032 | 0.150 ** | 0.047 | −0.308 ** | 0.028 | −0.048 | −0.051 | 0.001 | −0.062 * | 0.057 * | −0.225 ** | 1 | −0.206 ** | −0.438 ** |
| V30 | 0.173 ** | −0.036 | −0.061 * | −0.143 ** | 0.058 * | −0.129 ** | −0.039 | 0.075 ** | 0.009 | 0.021 | −0.008 | −0.106 ** | −0.094 ** | −0.206 ** | 1 | −0.183 ** |
| V31 | 0.344 ** | −0.045 | −0.117 ** | −0.229 ** | −0.011 | −0.275 ** | −0.069 * | 0.031 | 0.012 | −0.017 | 0.039 | 0.020 | −0.200 ** | −0.438 ** | −0.183 ** | 1 |

Note: ** indicates that correlation is significant at 0.01 level (2-tailed). * indicates that correlation is significant at 0.05 level (2-tailed).

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
