# Peer review of "Analysis of Factors Associated with Highway Personal Car and Truck Run-Off-Road Crashes: Decision Tree and Mixed Logit Model with Heterogeneity in Means and Variances Approaches"

_informatics, doi:10.3390/informatics10030066_

Round 1
Reviewer 1 Report
This paper focuses on highway personal car and truck run-off-road crashes, using decision tree(DT) and random parameters binary logit model(RPBL) approaches to analyze the influencing factors. My comments are as follows:
1. DT and RPBL approaches may be archaic. And they are used directly in this article without improvement. Moreover, the contributions of this study are unclear.
2. To strengthen the literature review, some recent studies on random parameters approach are suggested to be added. For example:
Temporal analysis of crash severities involving male and female drivers: A random parameters approach with heterogeneity in means and variances
Random-Parameter Bayesian Hierarchical Extreme Value Modeling Approach with Heterogeneity in Means and Variances for traffic conflict-based crash estimation
3. The results of model comparison are not obvious. Accuracy results, sensitivity results and specificity results of truck are very similar between DT and RPBL, which can not explain the advantages and disadvantages of the model.
4. There are some errors with the format in ‘Table’ and ‘References’.
5. Please add analysis of the significant impact factor in Table 3 results.
6. the practical applications of the findings from this study are advised to be added.
7. The randomized parameter setting method is not specified in Equation (1). Please add formulas for the randomization parameter.
Moderate editing of English language required.
Reviewer 2 Report
The paper titled "Analysis of factors associated with highway personal car and truck run-off-road crashes: Decision tree and random parameters logit model approaches" . Overall, paper quality can be improved; however, the following major revisions can be considered
1. Abstract needs to be rewritten well. It is necessary to highlight the study's main findings. Elaborate abstract in more detail in terms of research methodology, including data duration, and specific policy implication.
2. There is a lack of clarity of ideas in the introduction. It needs to be expanded further to cover all the critical aspects of the topic including sustainability and electronics car idea in introduction and literature review. Also, authors should enriched their literature review sections through highlights the importance of green aspects in automobile industry. Following articles can be cited for strengthen the scientific quality:
https://doi.org/10.1177/00368504221145648
https://doi.org/10.3390/su14084524
3. In the Introduction part, clearly demonstrate the major aim of this study in the last paragraph of introduction that what's novel in it and how it will contribute to the existing literature
4. Please carefully check recent literature and discuss/cite as you see fit, and update your reference list.
5. The conclusion section provides a lack of contributions to this manuscript. Provide the key features, merits, and limitations of the proposed approach to clarify the precise boundary of the algorithms. The implication of the proposed method is also required.
6. Please proofread the entire manuscript and remove the English structural and grammatical erros.
7. It is recommended to crosscheck and match to the entire citations in the article with reference list.
Language is understandable but needs to be improved through proofreading of entire draft.
Reviewer 3 Report
This paper utilizes Decision Tree and Random Parameters Logit models to analyze run-off-road crashes in Thailand, as well as the factors contributing to such incidents. Despite the topic's importance - particularly in Thailand - the models used, their performance, and the overall strength of the analysis appear relatively weak. Please find my detailed comments below:
The statement, "the accident statistics report also revealed that personal cars and trucks are among the most related to road crash damages" seems to contradict Figure 2, where trucks appear to be the category with the smallest percentage.
The Decision Tree is perhaps the simplest machine learning model. The authors should consider testing and comparing more complex ML models.
If a seperate literature review section is not presented, then the literature review in the introduction should be enhanced.
The authors do not appear to have checked for multicollinearity among the input parameters. Multicollinearity can result in unstable parameter estimates and should be taken into consideration.
The truck model has a significantly smaller AIC value compared to the car model. However, McFadden R2 values of 0.0593 and 0.0446 suggest the models explain a relatively small portion of the variance in the dependent variable, especially for the truck model. The authors should consider improving the model or at least provide explanations for these limitations.
The Sensitivity of both models for trucks is unacceptably low (0% and 1.49%). This serious issue suggests that both models are performing poorly in recognizing positive cases. The authors might need to adjust the parameters of their model, use a different model, or reevaluate the quality and relevance of their input data.
The model performance section should precede the analysis of contributing factors to crashes. Before interpreting the findings, the authors must first demonstrate the utility of the model.
A proofread of the paper would be beneficial.
Round 2
Reviewer 1 Report
All the reviewer comments have been addressed, and the paper is recommended for publication.
Only a minor English check is required.
Author Response
We express our gratitude for your invaluable insights. Your comments have greatly contributed to the enhancement of our manuscript. In terms of the language aspect, we have meticulously reviewed the entirety of the manuscript.
Reviewer 2 Report
Accepted
Author Response
We express our gratitude for your invaluable insights. Your comments have greatly contributed to the enhancement of our manuscript.
Reviewer 3 Report
The authors addressed most of my questions, and the revised paper has improved significantly. However, I have one remaining concern: after updating the mix logit model, the McFadden R² is still the same. Should it be different now that you are using RPBLHMV? Please double check. Nevertheless, a McFadden R² of 5% is still considered very low. Please see if an improvement can be provided.
One minor comment: 'In contrast, other DT models that utilize Blackbox algorithms cannot reveal the priority of factors.' I assume you mean other ML models here?"
The language is okay.
